Persistent homology classification algorithm

http://orcid.org/0000-0003-3883-7773 De Lara Mark Lexter D. 1 2 mddelara@up.edu.ph
1 Institute of Mathematical Sciences and Physics, College of Arts and Sciences, University of the Philippines Los Baños , College, Los Baños, Laguna , Philippines
2 Institute of Mathematics, University of the Philippines Diliman , Quezon City, Metro Manila , Philippines
Ventura Sebastian
Electronic publication date: 2023 Jan 10
Publication date: 2023
Volume: 9
Electronic Location ID: e1195
Received 2021 Aug 17; Accepted 2022 Dec 1
Copyright: © 2023 De Lara
Copyright year: 2023
Copyright holder: De Lara
License: This is an open access article distributed under the terms of the Creative Commons Attribution License, which permits unrestricted use, distribution, reproduction and adaptation in any medium and for any purpose provided that it is properly attributed. For attribution, the original author(s), title, publication source (PeerJ Computer Science) and either DOI or URL of the article must be cited.
License URL: https://creativecommons.org/licenses/by/4.0/

Keywords: Persistent homology, Supervised learning, Classification algorithm, Topological data analysis

Funding: University of the Philippines Los Baños Accelerated Science and Technology Human Resource Development Program of the Department of Science and Technology This work was supported by the University of the Philippines Los Baños and by the Accelerated Science and Technology Human Resource Development Program of the Department of Science and Technology. The funders had no role in study design, data collection and analysis, decision to publish, or preparation of the manuscript.

==============================
Data classification is an important aspect of machine learning, as it is utilized to solve issues in a wide variety of contexts. There are numerous classifiers, but there is no single best-performing classifier for all types of data, as the no free lunch theorem implies. Topological data analysis is an emerging topic concerned with the shape of data. One of the key tools in this field for analyzing the shape or topological properties of a dataset is persistent homology, an algebraic topology-based method for estimating the topological features of a space of points that persists across several resolutions. This study proposes a supervised learning classification algorithm that makes use of persistent homology between training data classes in the form of persistence diagrams to predict the output category of new observations. Validation of the developed algorithm was performed on real-world and synthetic datasets. The performance of the proposed classification algorithm on these datasets was compared to that of the most widely used classifiers. Validation runs demonstrated that the proposed persistent homology classification algorithm performed at par if not better than the majority of classifiers considered.

Introduction

Machine learning is an important branch of artificial intelligence that deals with the study of computer systems and algorithms that can learn and develop on their own without being explicitly programmed to do so. It is focused on the development of computer programs capable of performing data processing and predictive analysis. The three major categories of machine learning techniques are supervised, unsupervised, and reinforcement learning. Supervised learning is a process in which a system learns from a readily available training dataset of accurately classified observations. One of the major tasks addressed by supervised learning is classification.

Classification algorithms

Classification is the process of identifying, recognizing, grouping, and understanding new observations into categories or classes (Alpaydin, 2014). A training dataset is composed of ( n+1)-dimensional data points which are split into an n-dimensional input vector often called attributes and into a one-dimensional output category or class. A dataset can be univariate, bivariate, or multivariate in nature, whereas the attributes are quantifiable properties that can be categorical, ordinal, integer-valued, or real-valued. A classification algorithm or classifier is a procedure that performs classification tasks. The term classifier may also refer to the mathematical function that maps input attributes to an output category.

Classification algorithms have found a wide range of applications in domains such as computer vision, speech recognition, biometric identification, biological classification, pattern recognition, document classification, and credit scoring (Abiodun et al., 2018). Classification problems can be categorized as binary or multi-class. Binary classification entails allocating an observation to one of two distinct categories, whereas multi-class classification involves assigning an observation to one of more than two distinct classes.

Numerous classification algorithms have been developed since the rise of artificial intelligence. While several of these classifiers are specifically developed to solve binary classification problems, some algorithms can be used to solve binary and multi-class classification problems. Many of these multi-class classifiers are extensions or combinations of one or more binary classifiers.

The no-free lunch theorem proved by David Wolpert and William Macready implies that no learning or optimization algorithm works best on all given problems (Wolpert, 1996; Wolpert & Macready, 1997). A classifier can be chosen depending on the type of data at hand. Since then, many state-of-the-art classifiers have been developed. Some of the most commonly used classifiers are logistic regression, Naive Bayes classifier, linear discriminant analysis, support vector machines, k-nearest neighbor, decision trees, and neural networks.

A classification algorithm is a linear classifier if it uses a linear function or linear predictor that assigns a score to each category k based on the dot product of a weight vector and a feature vector. The linear predictor is given by the score function, Score(Xi,k)=βkXi, where Xi is the feature vector for the observation i, and βk is the weight vector corresponding to category k. Observation i is mapped by the linear predictor to the category k with the highest score function βkXi. Examples of linear classifiers include logistic regression, support vector machines, and linear discriminant analysis (Yuan, Ho & Lin, 2012).

Topological data analysis (TDA)

Data scientists employ techniques and theories drawn from various fields of mathematics, particularly algebraic topology, statistics, information science, and computer science. In mathematics, there is a growing field called topological data analysis, which is an approach that uses tools and techniques from topology to analyze datasets (Chazal & Michel, 2021). In the past two decades, TDA has been applied in various areas of science, engineering, medicine, astronomy, image processing, and biophysics.

Analyzing the shape of data is one of the motivations for using TDA. One of the key tools used in TDA is persistent homology (PH), a method based on algebraic topology that can be used for determining invariant features or topological properties of a space of points that persist across multiple resolutions (Edelsbrunner & Harer, 2008; Carlsson, 2009; Edelsbrunner & Harer, 2010). These derived invariant features are sensitive to small changes in the input parameters which make PH attractive to researchers studying qualitative features of data. PH involves representing a point cloud by a filtered sequence of nested complexes, which are turned into novel representations like barcodes and then interpreted statistically and qualitatively based on persistent topological features obtained (Otter et al., 2017). The following section gives a brief description of how to compute PH and a more detailed discussion of pertinent information about the homology of simplicial complexes and the complete process of computing PH of a point cloud can be found in the Appendix.

Computing persistent homology

Given a finite metric space S, such as a point cloud or a collection of experimental data, it is assumed that S represents a sample from an underlying topological space, such as space T. Describing the topology of T based only on a finite sample S is not a straightforward task. However, PH can be utilized to determine the general shape and topological properties of T while maintaining the robustness of the data.

A point cloud can undergo a filtration process that turns it into a sequence of nested simplicial complexes. This is accomplished by considering a finite number of increasing parameters and recording the sublevel sets that track changes in the topological features of the complexes. More specifically, gradually thickening the points give birth to new topological features, like simplices and n-dimensional holes (Zomorodian & Carlsson, 2005). Figure 1 shows an illustration of the filtration process. A more formal and extensive discussion of how to compute the persistent homology of a point cloud is presented in the Appendix.

Figure 1 Filtration of a point cloud into a nested sequence of simplicial complexes, with K1⊆K2⊆...⊆K5.

The appearance (birth) and disappearance (death) of intrinsic topological features, like homology groups and Betti numbers, can be recorded and visualized in many ways. The most frequently used are persistence barcodes or persistence diagrams. The duration or life span of the persisting topological features is essential in analyzing the qualitative and topological properties of a dataset.

Definition 1. Let K be a finite simplicial complex and K1⊆K2⊆…⊆Kr=K be a finite sequence of nested subcomplexes of K. K is called a filtered simplicial complex and the sequence {K1,K2,…} is called a filtration of K.

The homology of each of the subcomplexes can be computed. For each p, the inclusion maps Ki→Kj induce 𝔽2-linear maps ∂ij:Hp(Ki)→Hp(Kj) for all i,j∈{1,2,…,r} with i≤j. It follows from functoriality that ∂kj∘∂ik=∂ij for all i≤k≤j.

Definition 2. Let Ks be a subcomplex in the filtration of the simplicial complex K, or Ks be the filtered complex at time s, and Zks=Ker∂ks and Bks=Im∂k+1s be the k-th cycle group and boundary group of Ks, respectively. The k-th homology group of Ks is Hks=Zks/Bks=Ker(∂ks)/Im(∂k+1s).

Definition 3. For p∈{0,1,2,…}, the p-persistent k-th homology group of K given a subcomplex Ks is

Hks,p(K,Ks)=Hks,p(K)=Zks/(Bks+p⋂Zks)=Ker(∂ks)Im(∂k+1s+p)⋃Ker(∂ks).

The p-th persistent k-th Betti number βks,p of Ks is the rank of Hks,p(K).

Persistent vector spaces and persistence diagrams

Definition 4. A persistent vector space Carlsson (2014) is defined to be a family of vector spaces with linear maps between them, denoted by {Vδ⟶vδ,δ′Vδ′}δ≤δ′∈ℝ, such that (1) dim(Vδ)<∞ for each δ∈ℝ,

(2) there exist δI, δF∈ℝ such that all maps vδ,δ′ are linear isomorphisms for δ,δ′≥δF and for δ,δ′≤δI, and

(3) there are only finitely many values of δ∈ℝ such that vδ−ϵ≇vδ

(4) for any δ<δ′<δ′′∈ℝ, we have vδ′,δ′′∘vδ,δ′=vδ,δ′′.

Note that there are only finitely many values of δ∈ℝ for which the linear maps are not isomorphisms. This implies that the indexing set ℝ can be reduced to ℕ, and we may equivalently define a persistent vector space to be an ℕ-induced family {Vδi⟶vi,i+1Vi+1}i∈ℕ such that (1) dim(Vδi)<∞ for each i∈ℕ and

(2) there exists F∈ℕ such that all the vi,i+1 are isomorphisms for i≥F.

A persistent vector space can be obtained from a filtered simplicial complex and for each persistent vector space, one may associate a multiset of intervals called a persistence barcode or persistence diagram. A barcode represents each persistent generator with a horizontal line beginning at the first filtration level where it appears and ending at the filtration level where it disappears. Short bars correspond to noise or artifacts in the data, while lengthy bars correspond to relevant topological features. The standard treatment of persistence barcodes and diagrams appears in Edelsbrunner, Letscher & Zomorodian (2002) and Zomorodian & Carlsson (2005).

Presented here is a modern presentation of the definition of persistence diagrams and persistence barcodes (Chowdhury & Mémoli, 2018; Edelsbrunner, Jablonski & Mrozek, 2015).

Let V={Vδi⟶vi,i+1Vi+1}i∈ℕ be a persistent vector space. Compatible bases (Bi)i∈ℕ for each Vδi, i∈ℕ can be chosen such that vi,i+1|Bi is injective for each i∈ℕ, and rank(vi,i+1)=card(im((vi,i+1|Bi)∩Bi+1)), for each i∈ℕ (Edelsbrunner, Jablonski & Mrozek, 2015). Note that vi,i+1|Bi denotes the restriction of vi,i+1 to the set Bi. Fix such a collection (Bi)i∈ℕ of bases. Then, define

L:={(b,i)|b∈Bi,b∉Im(vi−1,i),i∈{2,3,…}}∪{(b,1)|b∈B1}.

For each (b,i)∈L, the integer i is interpreted as the birth index of the basis element b or the first index at which a certain algebraic signal appears in the persistent vector space. Now, define a map l:L→ℕ as follows

ℓ(b,i):=max{k∈ℕ|vk−1,k∘…∘vi+1,i+2∘vi,i+1(b)∈Bk}.

For each (b,i)∈L, the integer ℓ(b,i) is called the death index or the index at which the signal b disappears from the persistent vector space.

Definition 5. The persistence barcode of V is defined to be the following multisets of intervals:

Pers(V):=[[δi,δj+1)|there exists(b,i)∈L such that ℓk(b,i)=j],

where the bracket notation denotes taking the multiset and the multiplicity of [δi,δj+1] is the number of elements (b,i)∈L such that ℓ(b,i)=j.

The elements of Pers(V) are called persistence intervals. They can be represented as a set of line segments over a single axis. Equivalently, Pers(V) can be visualized as a multiset of points lying on or above the diagonal in ℝ¯2 counted with multiplicity, where ℝ¯ denotes [−∞,+∞].

Definition 6. The persistence diagram of V is given by

Dgm(V):=[(δi,δj+1)∈ℝ¯2|[δi,δj+1)∈Pers(V)],

where multiplicity of (δi,δj+1)∈ℝ¯2 is given by the multiplicity of [δi,δj+1)∈Pers(V).

Discussion of the robustness and stability of the persistence diagram requires the notion of distance. Given two persistence diagrams, say X and Y, the definition of distance between X and Y is given as follows.

Definition 7. Let p∈[1,∞]. The p-th Wasserstein distance between X and Y is defined as

Wp[d](X,Y):=infϕ:X→Y⁡[∑x∈Xd[x,ϕ(x)]p]1/p

for p∈[1,∞) and as

W∞[d](X,Y):=infϕ:X→Y⁡supx∈X⁡d[x,ϕ(x)]

for p=∞, where d is a metric on ℝ¯2 and ϕ ranges over all bijections from X to Y.

Normally, d is taken to be Lq where q∈[1,∞] and the most commonly used distance function is the Bottleneck distance W∞[L∞]. Equivalently, the bottle distance between persistence diagrams can be defined as follows.

Definition 8. The bottleneck distance between persistent diagrams and more generally, between multisets C, D of points in ℝ¯2 is given by

dB(C,D):=inf{supa∈C⁡‖a−ϕ(a)‖|ϕ:C∪Δ∞→D∪Δ∞ is a bijection.},

where ‖(p,q)−(p′,q′)‖∞:=max(|p−p′|,|q−q′|) for each p, q, p', and q′∈ℝ, and Δ∞ is the multiset consisting of each point on the diagonal, taken with infinite multiplicity.

Basis for the classification algorithm

Let I be an index set with K elements, where K∈ℕ. Suppose each k∈I corresponds to a class of points in ℝn. Let Xk be the set of n-dimensional points in the Euclidean space belonging to class k. Now, suppose there is an n-dimensional point αp=(α1p,α2p,…,αnp)∈ℝn. The problem of determining which of the K classes of points αp belongs to is a classification problem. Note that the collection of points belonging to the same class has its own topological properties. It is a natural intuition that the inclusion of an additional point αp in a point cloud where it belongs should result in a very small change in its topological properties. What follows is a way to analyze a point cloud and its properties.

Given a point cloud, we can compute its corresponding topological properties in the form of a persistence barcode or a persistence diagram. To do this, we follow the succeeding construction. From here on, this is referred to as the long construction.

For each point cloud Xk, k∈I, applying the long construction is the same as defining the following; (i) the Vietoris Rips complex filtration given by X1k⊆X2k⊆…⊆Xm+1k corresponding to the Xk with fixed m∈ℕ, ϵ>0, and the parameter values ϵj’s with ϵ1<ϵ2<…<ϵm+1 and ϵj=(j−1)ϵm, for each j∈{1,2,…,m+1},

(ii) an ℕ-indexed persistent vector space Xk={Xik⟶xi,i+1kXi+1k}i∈ℕ such that

(1) dim(Xik)<∞ for each i∈ℕ and

(2) there exists m∈ℕ such that all the xi,i+1 are isomorphisms for i≥m.

(iii) chosen compatible bases (Bik)i∈ℕ for each respective class k or Xik, k∈I such that xi,i+1k|Bik denote the restriction of xi,i+1k to the set Bik and rank(xi,i+1k)=card(im((xi,i+1|Bik)∩Bi+1)),

(iv) the set given by Lk:={(b,i)|b∈Bik,b∉im(xi−1,ik),i∈{2,3,…}}∪{(b,1)|b∈B1k},

(v) the map defined by ℓk(b,i):=max{r∈ℕ|xr−1,r∘…∘xi+1,i+2∘xi,i+1(b)∈Brk},

(vi) the persistence barcode given by Pers(Xk):=[[δi,δj+1)|there exists(b,i)∈ Lk such that ℓk (b,i)=j], where the bracket notation denotes taking the multiset and the multiplicity of [δi,δj+1] is the number of elements (b,i)∈Lk such that ℓk(b,i)=j, and

(vii) the persistence diagram given by Dgm(Xk):=[(δi,δj+1)∈ℝ¯2|[δi,δj+1)∈Pers(V)], where multiplicity of (δi,δj+1)∈ℝ¯2 is given by the multiplicity of [δi,δj+1)∈Pers(Xk).

Now, we consider the scenario of adding the n-dimensional point αp=(α1p,α2p,…,αnp)∈ℝn to the point clouds. Define Yk=Xk∪{αp} for each k∈I. Apply the long construction for each of these point clouds, that is define the following; (i) the Vietoris Rips complex filtration given by Y1k⊆Y2k⊆…⊆Ym+1k corresponding to the Yk with fixed m∈ℕ, ϵ>0, and the parameter values ϵj’s where ϵ1<ϵ2<…<ϵm+1 and ϵj=(j−1)ϵm for each j∈{1,2,…,m+1},

(ii) the ℕ-indexed persistent vector space Yk={Yik⟶yi,i+1kYi+1k}i∈ℕ similar to that of Xk,

(iii) compatible bases (Bi~k)i∈ℕ similar to that of (Bik)i∈ℕ,

(iv) the set L~k similar to that of Lk,

(v) the map ℓ~k similar to that of ℓk

(vi) the persistence barcode or multiset given by Pers(Yk), and

(vii) the persistence diagram given by Dgm(Yk).

Finally, classification is done by choosing the class k for which the change in the topological properties between Xk and Yk is minimum. This is inspired by the following observations.

Theorem 1. Let I be a finite index set. Suppose that for each k∈I, Xk is the set of points in ℝn belonging to class k, and that Xi∩Xj=∅ for each i≠j. Assuming αp=(α1p,α2p,…,αnp)∈ℝn is a point identical to one of the points in Xp, for an element p∈I. Then, for each k∈I,

dB(Dgm(Xp),Dgm(Yp))≤dB(Dgm(Xk),Dgm(Yk)),

where Xk, and Yk are the ℕ-indexed persistent vector spaces corresponding Xk and Xk∪{αp}, respectively, for each k∈I.

Proof. Let αp∈Xp be an n-dimensional point where p∈I. Choose any element k∈I such that k≠p. Then, Xk∩Xp=∅. Apply the long construction for Xk, Yk=Xk∪{αp}, Xp, and Yp=Xp∪{αp}. From the long constructions leading to the computation of persistence diagrams, we can define the following (i) the Vietoris Rips complex filtrations associated to Xk, Yk, Xp and Yp, with respect to some m∈ℕ, ϵ>0, and the parameter values ϵj’s with ϵ1<ϵ2<…<ϵm+1 and ϵj=(j−1)ϵm for each j∈{1,2,…,m+1},

(ii) the ℕ-indexed persistent vector spaces Xk, Yk, Xp, and Xp,

(iii) respective compatible bases (Bik)i∈ℕ, (Bip)i∈ℕ, (Bi~k)i∈ℕ, and (Bi~p)i∈ℕ,

(iv) the sets Lk, Lp, L~k, and L~p,

(v) the maps ℓk, ℓp, ℓ~k, and ℓ~p,

(vi) the persistence barcodes or multisets given by Pers(Xk), Pers(Yk), Pers(Xp), and Pers(Yp), and

(vii) the persistence diagrams given by Dgm(Xk), Dgm(Yk), Dgm(Xp), and Dgm(Yp).

Since αp∈Xp, then αp∉Xk, Xp=Yp, and Yk has one more element than Xk. Also, αp∈B1~k and αp∉B1k. Note that there is at least one (αp,i)∈Lk~ with i at least the first index 1. Then ℓk~(αp,i)≥1, by the definition of ℓk~:Lk~→ℕ. That is, the death index corresponding (αp,i) must be greater than 1.

This implies that Pers(Yk) will have one more element than Pers(Xk). This means that Pers(Yk) has one unmatched point with respect to the pairing with Pers(Xk). Then, the computation of dB(Dgm(Yk),Dgm(Xk)) boils down to the optimal distance between the matched points or the distance between the unmatched point to the diagonal. In either case, this is greater than zero.

Now, since Xp=Yp, then dB(Dgm(Yp),Dgm(Xp))=0.

Hence, for each k∈I, dB(Dgm(Xk),Dgm(Yk))≥dB(Dgm(Xp),Dgm(Yp))=0.

Theorem 2. Let I be a finite index set. Suppose that for each k∈I, Xk is the set of points in ℝn belonging to class k, and that Xi∩Xj=∅ for each i≠j. Assume αp∈ℝn is the new data point and αp∉Xk for each k∈I. Let αq∗ be a point in Xq∗, where q∗∈I, such that αq∗ is the closest point to αp among all points in ⋃k=1KXk. If the long construction is applied to Xq∗ and Yq∗=Xq∗∪{αp}, and to Xk and Yk=Xk∪{αp}, for each k∈I, k≠q∗, and the sets Lk, Lq∗, L~k, and L~q∗ are limited to elements with birth index of 1, then for each k∈I,

dB(Dgm(Xq∗),Dgm(Yq∗))≤dB(Dgm(Xk),Dgm(Yk)),

where Xk and Yk are the ℕ-indexed persistent vector spaces corresponding to Xk and Xk∪{αp}, respectively, for each k∈I.

Proof. Let I be a finite index set. Suppose that for each k∈I, Xk is the set of points in ℝn belonging to class k, and that Xi∩Xj=∅ for each i≠j. Assume αp∈ℝn is the new point and αp∉Xk for each k∈I. For each k∈I, let αkq be the element in Xk which is closest to αp. Suppose that αq∗ is the closest element to αp among any αkq, where k∈I.

Without loss of generality, choose any element k∈I such that k≠q∗. Then, Xk∩Xq∗=∅. Apply the long construction for Xk, Yk=Xk∪{αp}, Xq∗ and Yq∗=Xq∗∪{αp}. From the long constructions leading to the computation of persistence diagrams, we define the following; (i) the Vietoris Rips complex filtrations associated to Xk, Yk, Xq∗and Yq∗, with respect to some m∈ℕ, ϵ>0, and the parameter values ϵj’s with ϵ1<ϵ2<…<ϵm+1 and ϵj=(j−1)ϵm, for each j∈{1,2,…,m+1},

(ii) the ℕ-indexed persistent vector spaces Xk, Yk, Xq∗, and Xq∗,

(iii) respective compatible bases (Bik)i∈ℕ, (Biq∗)i∈ℕ, (Bi~k)i∈ℕ, and (Bi~q∗)i∈ℕ,

(iv) the sets Lk, Lq∗, L~k, and L~q∗ whose elements are restricted to those with birth index of 1,

(v) the maps ℓk, ℓq∗, ℓ~k, and ℓ~q∗,

(vi) the persistence barcodes or multisets given by Pers(Xk), Pers(Yk), Pers(Xq∗), and Pers(Yq∗), and

(vii) the persistence diagrams given by Dgm(Xk), Dgm(Yk), Dgm(Xq∗), and Dgm(Yq∗).

For each k∈I, define αkq∈ℝn to be the point in Xk which is closest to αp. Note that limiting the basis elements to those with birth index of 1 means that only the 0-dimensional topological properties or the connectedness of the simplices are considered. Also, for each k∈I, Xk and Yk differ only by a single point, the point αp. By these, the 0-dimensional hole corresponding to αp in the filtration of Yk disappears faster if αkq is closer to αp. Moreover the computation of dB(Dgm(Xk),Dgm(Yk)) is reduced to the computation of ℓ~k(αp,1)=12‖αP−αkq‖ for each k∈I. Since αq∗ is closest to αp among all the αkq’s, then this implies that ℓ~q∗(αq∗,1)≤ℓ~k(αkq,1), for any k∈I, where (αq∗,1)∈(Bi~q∗)i∈ℕ and (αkq,1)∈(Bi~k)i∈ℕ. This completes the proof.

Illustration 1. Consider the point clouds labelled as X1, X2, and X3, and the point P in the 2-dimensional space as presented in Fig. 2. Suppose that the point P has to be categorized into one of the point clouds.

Figure 2 Point clouds X1, X2, and X3, and point P.

For each i∈{1,2,3}, define Yi=Xi∪{P} and compute the persistent homologies of Xi and Yi. Presented in Fig. 3 are the respective persistence barcodes corresponding to the 0-dimensional holes in the filtration of the Xi’s and Yi’s. Note that each pair of Xi and Yi differ only by a single point, the point P. Since only the 0-dimensional holes are considered here, then the persistence barcode for Yi has one more bar than the persistence barcode for Xi, for each i∈{1,2,3}. These differences, represented by the green bars in Fig. 3, are indicative of the shortest distance of P from the point clouds. The life span of the 0-dimensional hole corresponding to point P in the filtration of Yi is the shortest when Xi is the closest point cloud to P. For each k∈{1,2,3},

Figure 3 Persistence barcodes corresponding to the 0-dimensional holes in the filtration of the Xi’s and Yi’s.

dB(Dgm(X3),Dgm(Y3))≤ℓ~k(P,1)=12‖P−αkq‖,

where αkq is the point in Xk that is closest to P.

Advances in the fusion of persistent homology and machine learning

Computation of persistent homology (PH) has been applied to a variety of fields, including image analysis, pattern comparison and recognition, network analysis, computer vision, computational biology, oncology, and chemical structures. Some examples can be found in the works of Charytanowicz et al. (2010), Goldenberg et al. (2010), Nicolau, Levine & Carlsson (2011), Xia & Wei (2014), Giansiracusa, Giansiracusa & Moon (2019), and Ignacio & Darcy (2019). Furthermore, advances in the different aspects of computing PH have been increasing at a very rapid rate. Various software have also been developed for computing persistent homology. These software packages include JavaPlex, Perseus, Dipha, Dionysus, jHoles, GUDHI (Geometry Understanding in Higher Dimensions), Rivet, Ripser, PHAT (Persistent Homology Algorithm Toolkit), and R-TDA (R package for Topological Data Analysis) (Otter et al., 2017; Pun, Lee & Xia, 2022). On the other hand, machine learning has been in development as early as 1950s, but it was not until the 1990s that a shift from a knowledge-driven approach to a data-driven approach in studying machine learning took place. It was also during this time when support vector machines and neural networks became very popular. Persistent homology, which has only been around for a decade, has also received attention in recent years.

A direct exposition of the use of machine learning and persistence barcodes was used by Giansiracusa, Giansiracusa & Moon (2019) in solving a fingerprint classification problem. They also showed that better accuracy rates can be achieved when applying topological data analysis to 3-dimensional point clouds of oriented minutiae points. Chung, Cruse & Lawson (2020) used persistence curves, rather than barcodes or diagrams, to analyze time series classification problems. Ismail et al. (2020) used PH-based machine learning algorithms to predict next day direction of stock price movement. Hofer et al. (2017) incorporated topological signatures to deep neural networks and performed classification experiments on 2D object shapes and social network graphs. Gonzalez-Diaz, Gutiérrez-Naranjo & Paluzo-Hidalgo (2020) used PH to define neural networks by incorporating the concept of representative datasets. In their study, they used persistence diagrams in choosing points that would be included in the representative dataset. Chen et al. (2019) leveraged topological information to regularize the topological complexity of kernel classifiers by incorporating a topological penalty, while Pokorny, Hawasly & Ramamoorthy (2014) used topological approaches to improve trajectory classification in robotics. Edwards et al. (2021) introduced TDAExplore, a machine learning image analysis pipeline based on topological data analysis. TDAExplore can be used to classify high-resolution images and characterize which image regions contribute to classification. PH has also been found useful in unsupervised learning and clustering tasks. Islambekov & Gel (2019) presented an unsupervised PH-based clustering algorithm for space-time data. They evaluated the performance of their proposed algorithm on synthetic data and compared it to other well-known clustering algorithms such as K-means, hierarchical clustering, and DBSCAN (Density-based spatial clustering of applications with noise).

Pun, Lee & Xia (2022) published a survey of PH-based machine learning algorithms and their applications. They presented ways how to use persistent homologies to improve machine learning algorithms such as support vector machines, tree-based methods, and artificial neural networks. The strategies require extracting topological features from individual data points and adding them as new attributes to the data points. The classification algorithms are then modified in such a way that they can utilize the additional attributes to classify the data points more accurately. These strategies of modifying learning algorithms by incorporating topological signatures as additional features to the data points are very effective when the data under scrutiny contains high-dimensional points and when points belonging to the same class share common topological properties. This means that the improved algorithm can be used to group together data points with similar shapes or topological properties. On the other hand, these strategies may not work when the dataset under consideration contains data points in a Euclidean space, when each of the points in the dataset has very low dimensions, or if data points from the same class do not share the same topological properties.

On the contrary, the objective of this study was to develop a supervised classification algorithm for dealing with classification problems involving points in a Euclidean space, datasets that can be separated into classes using hyperplanes, or when points from the same class have a certain level of proximity with each other. The proposed classifier in this study takes substantial use of persistent homologies and topological signatures associated with the different classes of points in the dataset instead of the individual data points’ persistent homologies and topological features.

Persistent homology classification algorithm (phca)

A supervised classification algorithm or classifier is a technique that assigns a new object or observation to a class based on the training set and the new observation’s attribute values. The classifier proposed in this work, termed persistent homology classification algorithm (PHCA), is largely reliant on the topological properties of each class of points in the available training set. The topological features derived from the persistence diagrams, generated by computing the persistent homology of each class in the training set, will be used to construct a linear classifier or a score function for classifying new observations. The section titled Computing Persistent Homology contains a discussion of the basis of this algorithm, while this section shows the outline of the proposed algorithm’s implementation.

Interpretation of a point cloud’s persistent homology

Let Z be a point cloud. Computing the persistent homology of Z means letting Z undergo filtration and recording the topological properties of Z that persist across multiple resolutions. The result of computing the persistent homology of Z or the summary of the topological properties of the space underlying Z can be visualized using persistence barcodes and persistence diagrams as shown in Definitions 5 and 6. See the Appendix for an illustration. Moreover, it can also be represented by an ni×3 matrix and it will be denoted by P(Z) in this section. The number of rows of P(Z), ni, is the number of topological features or n-dimensional holes detected in the filtration of Z. The 0-dimensional holes are the connected components, 1-dimensional holes refer to loops, 2-dimensional holes represent voids, and so on. One can limit the maximum dimension that can be detected in the filtration of Z. The first column entry of the i-th row of P(Z) indicates the dimension of i-th topological feature in the filtration of Z. The second and third column entries in the i-th row of P(Z) give the birth and death time of the i-th topological feature, respectively. The birth and death times refer to the filtration steps when the topological feature appears and disappears, respectively.

Let P(Z) be the persistent homology of a point cloud Z. Although it can be visualized as a persistence diagram, it denotes an ni×3 matrix, where ni is the number of holes, of various dimensions, which are detected in the complex filtration.

Setting of maximum parameter value

The maximum scales, denoted by maxsc, must be fixed. The scale here refers to the size of the parameter values that will be used in the filtration and the computation of the persistent homology of a point cloud. The suitable value for maxsc depends on the data at hand and it can be chosen to be equal to half of the maximum distance between any two points in the point cloud.

Training and classification

Assume that X is a training dataset consisting of ( n+1)-dimensional data points categorized into k classes. Each data point’s initial n entries contain its attributes, while the ( n+1)-th entry gives its label or classification. Suppose that X=X1∪X2∪…∪Xk, where X1,…, Xk are the k classes of data points, and mi is the number of elements in Xi. Each training point cloud Xi can be viewed as an ( mi) × ( n+1) matrix with each row representing a point belonging in Xi.

Suppose that a new n-dimensional point, denoted by α, needs to be classified. To begin, analyze the different classes in the training set or describe the topological properties of each class in X by computing P(Xi) for each i∈{1,…,k}. Then, measure the effect of including p in each class in X. Define Yi=Xi∪{α} and compute P(Yi) for each i∈{1,…,k}. Record the changes in the persistent homology between Xi and Yi for each i∈{1,…,k}. Specifically, record the change from P(Xi) to P(Yi) for each i∈{1,…,k}. Lastly, identify the class that is least impacted by the inclusion of α. This procedure follows the long construction defined in the basis for the classification algorithm.

Score function for PHCA

The use of persistent homology in the development of the score function that will act as the linear predictor for PHCA is described here. Consider a training set categorized into two classes, A and B. Suppose α is a point that needs to be classified and that α belongs to either A or B only, as shown in Fig. 4. The proposed PHCA works by computing the persistent homology of A, B, A∪{α}, and B∪{α}. Afterward, measure the changes in the topological features from A to A∪{α} and from B to B∪{α}. Suppose that point α is closer to point cloud A than point cloud B, then the change in the persistent homology of A to A∪{α} will be lower than the change in the persistent homology from B to B∪{α}. This is because the closer a point α to a point cloud, say Z, the birth, and death of some topological features of Z∪{p}, particularly the 0-dimensional holes, will come earlier. That is, the early appearance or disappearance of topological features translates to a shorter life span of topological features in the filtration of the complexes. This procedure which is based on Theorems 2 and 3 works particularly well when the point clouds for different classes are disjoint from one another or when a point under consideration is close to a particular point cloud. In light of these, the score function used for PHCA is computed in the following manner.

Figure 4 Training set of points categorized into class A (blue points) and class B (green points), and a new point q.

Recall that we have defined Yi=Xi∪{α} for each i∈{1,…,k}, and that P(Yi) is the persistent homology of Yi for each i∈{1,…,k}. The score function Score(Yi) is computed as the difference between the sum total of the lifespan of all the topological features in P(Yi) and the sum total of the lifespan of all the topological features in P(Xi). This is equivalent to the difference between the sum of the entries of the third column of P(Xi) and the sum of the entries of the third column of P(Yi). More explicitly, the score function value is computed as

(1) Score(Xi)=|∑a∈Yiℓ(a,1)−∑b∈Xiℓ(b,1)|,

where ℓ is the map that sends a basis element with birth index of 1 to its death index.

Finally, the new data point α is classified under class j if Score(Xj)≤Score(Xi) for all i∈{1,…,k}. Algorithm 1 gives the pseudocode for the persistent homology classification algorithm (PHCA).

Algorithm 1 Persistent Homology Classification Algorithm.

Require: X1,X2,…Xk,maxd,maxsc,andα	
Ensure: Class(α)	
 Procedure	
  for i=1tok do	
    P(Xi)←ni×3 matrix, a result of computing PH of Xi	
    P(Yi=Xi∪{α})←ni×3matrix,aresultofcomputingPHofYi	
   Compute Score (Xi)	
  end for	
   Class(α)←argmin∀i⁡{Score(Xi)}	
 end procedure	

Otter et al. (2017) established that computing the persistent homology of a point cloud using the standard algorithm has cubic complexity in terms of the number of simplices in the worst-case scenario. This, combined with the fact that PHCA is a linear classifier, means that the proposed classifier can be used to perform classification tasks in polynomial time.

Evaluvation methodolgy

The validity of PHCA was determined by solving four classification tasks using three well-studied datasets and a synthetic dataset made of points from three separate geometric figures. The number of classes per dataset ranges from 2 to 10, while the number of attributes per dataset ranges from 2 to 2025. The performance of the proposed PHCA was quantified and then compared to the performance of five main classification algorithms on four validation datasets. Linear discriminant analysis (LDA), Classification and Regression Trees (CART), K-Nearest Neighbors (KNN), Support Vector Machine (SVM), and Random Forest (RF) were the five classifiers used as benchmarks in this study.

All validation runs implemented in this study employed five-fold cross-validation. That is, the validation process involved dividing each class of the datasets into five sections. Each validation run requires a training set and a testing set. At any particular run, the testing set is composed of a partition from each class and the training set is composed of all the remaining data points. This guarantees that each class will have a representation in the testing. This scheme also assured that every single data point served as a testing point in one of the validation runs. To evaluate the methods, the confusion matrix was generated to determine the following metrics: precision, recall (sensitivity), specificity, accuracy, and F1 score. The confusion matrix gives the number of data points per class that is correctly predicted or incorrectly predicted.

For instance, consider a particular class, say Ci, among k classes. Then, we can define the following for each i∈{1,2,…,k}.

TPi is the number of true positives in class Ci, or the number of instances in Ci which are predicted to belong in Ci.

TNi is the number of true negatives in class Ci, or the number of instances outside Ci which are predicted to not belong in Ci.

FPi is the number of false positives in class Ci, or the number of instances outside Ci which are predicted to belong in Ci.

FNi is the number of false negatives in class Ci, or the number of instances in Ci which are predicted to not belong in Ci.

The five metrics per class Ci are computed as follows

(2) Precision for Class Ci :Prec(Ci)=TPiTPi+FPi

(3) Recall for Class Ci:Rec(Ci)=TPiTPi+FNi

(4) Specificity for Class Ci:Spec(Ci)=TNiTNi+FPi

(5) Accuracy for Class Ci:Acc(Ci)=TPi+TNiTPi+TNi+FPi+FNi

(6) F1 score for Class Ci:F1(Ci)=2×Prec(Ci)×Rec(Ci)Prec(Ci)+Rec(Ci)

A high sensitivity prediction in Class Ci implies that the reliability of predicting that an instance does not belong to Ci is high. However, predicting that an instance belong to Ci with high sensitivity is inconclusive. On the other hand, the high specificity of prediction in Class Ci implies that the reliability of predicting that an instance belongs to Ci is high. And, predicting that an instance does not belong to Ci with high specificity is inconclusive. Moreover, precision for Class Ci gives the proportion of data points predicted to really belong in Ci. Recall gives the proportion of data points in Class Ci which are correctly classified. Lastly, F1 score for Class Ci is a performance measure that combines recall and precision. F1 score is computed as the harmonic mean of precision and recall. This is a particular instance of the F β score which allows for more weight towards one of precision or recall over the other, needed for particular problems. An F-score may have a maximum value of 1.0, indicating perfect precision and recall, or a minimum value of 0, indicating that either precision or recall is zero.

The performance of PHCA and the five benchmark classification algorithms were compared using the Nemenyi post-hoc test. Pairwise comparisons of the methods were measured using mean rank differences. A p−value was computed for each pair of methods using the formula

=R¯i−R¯jk(n+1)12,

where Ri and Rj are mean ranks of two methods, k is the number of methods and n is the number of performance metric means per method. A p−value of less than α=0.05 implies that the two methods are significantly different.

The validation of PHCA and the benchmark classifiers were done in R using the TDA, Caret, and DescTools packages. R-TDA package, containing the Dionysius library, was used for the computation of the persistent homology of the point clouds. The Caret package was used for the implementation of the benchmark classifiers, with the same caret parameters, which are metric and control. While DescTools was used for the implementation of the Nemenyi Test.

Results and discussion

The performances of the proposed PHCA and the five benchmark classification algorithms (i.e., LDA, CART, KNN, SVM, and RF) for the four classification problems are presented here. Validation results for each classification task are also discussed in the following sections. Program codes, written in R, to implement the classification algorithms can be found on https://github.com/mlddelara/PHCA. Recall that PHCA works in a way that a data point in the testing set will be classified under a class if the inclusion of the point in the particular class’ training set results in the least change in the persistence diagram of the training set with the additional data point and least change in the engineered features corresponding the data point to be classified.

Iris plant dataset

The iris plant dataset composed of 150 observations created by Fisher (1936) was obtained from the UCI Machine Learning Repository (Dua & Graff, 2017). This is one of the commonly used datasets in pattern recognition. The dataset is divided into three categories: iris setosa, iris versicolour, and iris virginica. Each category is comprised of 50 data points, with four attributes namely sepal length, sepal width, petal length, and petal width (expressed in centimeters).

Table 1 shows the performance of PHCA and the five major classification algorithms in terms of precision and recall per class. Moreover, Table 2 shows the mean performance of the six methods in terms of accuracy, specificity per class, and F1 score per class. Barplots of these results are presented in Fig. 5. LDA, KNN, and PHCA got an accuracy of 96.67%, while CART, SVM and RF obtained an accuracy of 93.33%. Table 3 shows a summary of the Nemenyi test results and the p−values implies that there is no significant difference in the performance of PHCA and the other methods in terms of the mean performance metrics.

Table 1 Precision and recall per class of each of the six methods when classifying the iris dataset.

Classifier	Precision (%)	Recall (%)	
	Class 1	Class 2	Class 3	Class 1	Class 2	Class 3	
LDA	100.00	90.91	100.00	100.00	100.00	90.00	
CART	100.00	83.33	100.00	100.00	100.00	80.00	
KNN	100.00	90.91	100.00	100.00	100.00	90.00	
SVM	100.00	83.33	100.00	100.00	100.00	80.00	
RF	100.00	83.33	100.00	100.00	100.00	80.00	
PHCA	100.00	94.00	96.00	100.00	95.92	94.11	

Table 2 Accuracy, specificity per class, and F1 score per class of each of the six methods when classifying the iris dataset.

Classifier	Accuracy (%)	Specificity (%)	F1 score (%)	
		Class 1	Class 2	Class 3	Class 1	Class 2	Class 3	
LDA	96.67	100.00	95.00	100.00	100.00	95.24	94.74	
CART	93.33	100.00	90.00	100.00	100.00	90.91	88.89	
KNN	96.67	100.00	95.00	100.00	100.00	95.24	94.74	
SVM	93.33	100.00	90.00	100.00	100.00	90.91	88.89	
RF	93.33	100.00	90.00	100.00	100.00	90.91	88.89	
PHCA	96.67	100.00	97.03	97.98	100.00	94.95	95.05	
Number of data points:	150	Number of classes:	3		
Training set size:	120	Number of attributes:	4		
Testing set size:	30						

Figure 5 Barplots of performance metrics of PHCA and the five other classifiers for the iris dataset.

Table 3 Nemenyi’s test of pairwise comparison of the different classification algorithms for the iris dataset.

	Mean rank difference	p-value	
CART-LDA	−6.576923	0.9770	
KNN-LDA	0.000000	1.0000	
SVM-LDA	−6.576923	0.9770	
RF-LDA	−6.576923	0.9770	
PHCA-LDA	−3.346154	0.9990	
KNN-CART	6.576923	0.9770	
SVM-CART	0.000000	1.0000	
RF-CART	0.000000	1.0000	
PHCA-CART	3.230769	0.9992	
SVM-KNN	−6.576923	0.9770	
RF-KNN	−6.576923	0.9770	
PHCA-KNN	−3.346154	0.9990	
RF-SVM	0.000000	1.0000	
PHCA-SVM	3.230769	0.9992	
PHCA-RF	3.230769	0.9992	

Wheat seeds dataset

The wheat seeds dataset was created by Charytanowicz et al. (2010) at the Institute of Agrophysics of the Polish Academy of Sciences in Lublin. It is available at the UCI Machine Learning Repository (Dua & Graff, 2017). The dataset is composed of 210 observations, which is divided equally into three categories: kama, rosa, and canadian wheat variety. Each observation is characterized by seven attributes, namely, area, perimeter, compactness, length of kernel, width of kernel, asymmetry coefficient, and length of kernel groove. All of these attributes are real-valued and continuous.

The performance of PHCA and the five major classification algorithms in terms of precision and recall per class is shown in Table 4. Furthermore, Table 5 compares the six methods in terms of accuracy, specificity per class, and F1 score per class. Barplots of these results are presented in Fig. 6. For the wheat seeds dataset, PHCA got the third highest accuracy, which is at 90.95%. Notice that PHCA got the least performance in some metrics per class, but these were offset when PHCA got the highest performance in terms of precision for class 3, recall for class 1, specificity for class 3, and F1 score for class 3. Moreover, the results of the Nemenyi test, which can be found in Table 6, show that there is no significant difference between the performance of the six methods in terms of the mean performance metrics.

Table 4 Precision and recall per class of each of the six methods when classifying the wheat seeds dataset.

Classifier	Precision (%)	Recall (%)	
	Class 1	Class 2	Class 3	Class 1	Class 2	Class 3	
LDA	100.00	100.00	82.35	78.57	100.00	100.00	
CART	100.00	100.00	77.78	71.43	100.00	100.00	
KNN	100.00	93.33	73.68	57.14	100.00	100.00	
SVM	100.00	100.00	82.35	78.57	100.00	100.00	
RF	100.00	100.00	77.78	71.43	100.00	100.00	
PHCA	85.71	92.86	94.28	86.96	94.21	91.67	

Table 5 Accuracy, specificity per class, and F1 score per class of each of the six methods when classifying the wheat seeds dataset.

Classifier	Accuracy (%)	Specificity (%)	F1 score (%)	
		Class 1	Class 2	Class 3	Class 1	Class 2	Class 3	
LDA	92.85	100.00	100.00	89.3	88.00	100.00	90.30	
CART	90.48	100.00	100.00	85.70	83.30	100.00	87.50	
KNN	85.71	100.00	96.40	82.10	72.70	96.60	84.90	
SVM	92.86	100.00	100.00	89.30	88.00	100.00	90.30	
RF	90.48	100.00	100.00	85.70	83.30	100.00	87.50	
PHCA	90.95	92.90	96.50	97.10	86.30	93.50	93.00	
Number of data points:	210	Number of classes:	3		
Training set size:	168	Number of attributes:	7		
Testing set size:	42						

Figure 6 Barplots of performance metrics of PHCA and the five other classifiers for the wheat seeds dataset.

Table 6 Nemenyi’s test of pairwise comparison of the different classification algorithms for the wheat seeds dataset.

	Mean rank difference	p-value	
CART-LDA	−3.192308	0.9992	
KNN-LDA	−10.961538	0.8206	
SVM-LDA	0.000000	1.0000	
RF-LDA	−3.192308	0.9992	
PHCA-LDA	−9.653846	0.8871	
KNN-CART	−7.769231	0.9527	
SVM-CART	3.192308	0.9992	
RF-CART	0.000000	1.0000	
PHCA-CART	−6.461538	0.9787	
SVM-KNN	10.961538	0.8206	
RF-KNN	7.769231	0.9527	
PHCA-KNN	1.307692	1.0000	
RF-SVM	−3.192308	0.9992	
PHCA-SVM	−9.653846	0.8871	
PHCA-RF	−6.461538	0.9787	

Social network ads dataset

The social network ads dataset was created by Raushan (2017) and was retrieved from the Kaggle repository. The dataset is composed of 400 data points and is comprised of uneven number of observations per class. There are 143 data points for Class 1 and 257 data points for Class 2. Each data point has two attributes, age and estimated salary, and a class label, whether a customer purchased a product or not.

Table 7 shows the performance of PHCA and the five major classification algorithms in terms of precision, recall, specificity, F1 score, and accuracy. Barplots of these results are presented in Fig. 7. For this binary classification problem, PHCA got the highest recall at 86.89%. PHCA also got the second-highest F1 score and accuracy, which are at 88.55% and 85%, respectively. The results of the Nemenyi test, which is summarized in Table 8, show that there is no significant difference between the performance of PHCA and the other classification algorithms in terms of the mean performance metrics. Moreover, the mean rank difference of 18 and the p−value of 0.0155 between KNN and SVM shows that SVM performed significantly better than KNN.

Table 7 Precision, recall, specificity, F1 score, and accuracy (in percentage) of each of the six methods when classifying the social network ads dataset.

Classifier	Precision	Recall	Specificity	F1 Score	Accuracy	
LDA	80.77	82.35	64.28	81.55	75.95	
CART	95.35	80.39	92.85	87.23	84.81	
KNN	76.36	82.35	53.57	79.24	72.15	
SVM	95.56	84.31	92.85	89.58	87.34	
RF	90.91	84.31	85.71	84.21	81.01	
PHCA	90.27	86.89	81.20	88.55	85.00	
Number of data points:	400	Number of classes:	2	
Training set size:	181	Number of attributes:	2	
Testing set size:	119				

Figure 7 Barplots of performance metrics of PHCA and the five other classifiers for the social network ads dataset.

Table 8 Nemenyi’s test of pairwise comparison of the different classification algorithms for the social ads dataset.

	Mean rank difference	p-value	
CART-LDA	12.4	0.2252	
KNN-LDA	−2.2	0.9988	
SVM-LDA	15.8	0.0517	
RF-LDA	7.1	0.7987	
PHCA-LDA	11.3	0.3254	
KNN-CART	−14.6	0.0919	
SVM-CART	3.4	0.9903	
RF-CART	−5.3	0.9328	
PHCA-CART	−1.1	1.0000	
SVM-KNN	18.0	0.0155	
RF-KNN	9.3	0.5515	
PHCA-KNN	13.5	0.1477	
RF-SVM	−8.7	0.6234	
PHCA-SVM	−4.5	0.9661	
PHCA-RF	4.2	0.9749	

PHCA ranked 4th on all metrics with precision of 90.27%, recall of 86.89%, specificity of 81.2%, F1 score of 88.55%, and accuracy of 85%. PHCA bested LDA and KNN in terms of precision, specificity, F1 score, and accuracy, and it bested CART and RF in terms of recall, F1 score, and accuracy.

Synthetic dataset

This dataset was created by uniformly sampling 200 points from each of the following figures, the circle defined by x2+y2=25, the sphere defined by x2+y2+(z−1)2=1, and the torus defined by (3−x2+y2)2+(z+1)2=1. The x, y, and z coordinates of the 600 points served as the attributes, and the category was assigned according to which figure the points belong to.

Table 9 shows the performance of PHCA and the five major classification algorithms in terms of precision and recall per class. Moreover, Table 10 shows the performance of the six methods in terms of accuracy, specificity per class, and F1 score per class. Barplots of these results are presented in Fig. 8. For this synthetic dataset, PHCA and all benchmark classifiers, except LDA, got 100% for all performance metrics considered in this study. This is consistent with the results of the Nemenyi test which are summarized in Table 11. The p−values show that the performance of LDA is significantly different than the performance of the other algorithms, including PHCA.

Table 9 Precision and recall per class of each of the six methods when classifying the synthetic dataset.

Classifier	Precision (%)	Recall (%)	
	Class 1	Class 2	Class 3	Class 1	Class 2	Class 3	
LDA	76.92	96.55	100.00	100.00	70.00	97.5	
CART	100.00	100.00	100.00	100.00	100.00	100.00	
KNN	100.00	100.00	100.00	100.00	100.00	100.00	
SVM	100.00	100.00	100.00	100.00	100.00	100.00	
RF	100.00	100.00	100.00	100.00	100.00	100.00	
PHCA	100.00	100.00	100.00	100.00	100.00	100.00	

Table 10 Accuracy, specificity per class, and F1 score per class of each of the six methods when classifying the synthetic dataset.

Classifier	Accuracy (%)	Specificity (%)	F1 score (%)	
		Class 1	Class 2	Class 3	Class 1	Class 2	Class 3	
LDA	89.16	85.00	98.75	100.00	86.95	81.15	98.73	
CART	100.00	100.00	100.00	100.00	100.00	100.00	100.00	
KNN	100.00	100.00	100.00	100.00	100.00	100.00	100.00	
SVM	100.00	100.00	100.00	100.00	100.00	100.00	100.00	
RF	100.00	100.00	100.00	100.00	100.00	100.00	100.00	
PHCA	100.00	100.00	100.00	100.00	100.00	100.00	100.00	
Number of data points:	600	Number of classes:	3		
Training set size:	480	Number of attributes:	3		
Testing set size:	120						

Figure 8 Barplots of performance metrics of PHCA and the five other classifiers for the synthetic dataset.

Table 11 Nemenyi’s test of pairwise comparison of the different classification algorithms for the synthetic dataset.

	Mean rank difference	p-value	
CART-LDA	30	0.0096	
KNN-LDA	30	0.0096	
SVM-LDA	30	0.0096	
RF-LDA	30	0.0096	
PHCA-LDA	30	0.0096	
KNN-CART	0	1.0000	
SVM-CART	0	1.0000	
RF-CART	0	1.0000	
PHCA-CART	0	1.0000	
SVM-KNN	0	1.0000	
RF-KNN	0	1.0000	
PHCA-KNN	0	1.0000	
RF-SVM	0	1.0000	
PHCA-SVM	0	1.0000	
PHCA-RF	0	1.0000	

MNIST database of handwritten digits

The MNIST database (Modified National Institute of Standards and Technology database (Lecun et al., 1998)) is a massive library of handwritten digits that is often used for training and testing image processing techniques. It was made by “re-mixing” samples from the original datasets from NIST. The MNIST database of handwritten digits contains 60,000 training examples and 10,000 test instances. It is a subset of a bigger set accessible from the National Institute of Standards and Technology (NIST). The digits have been centered and size-normalized in a fixed-size picture. The original NIST black and white photos were resized to fit within a 20 × 20 pixel frame while maintaining their aspect ratio. The generated photos feature grey levels as a result of the normalization algorithm’s anti-aliasing method. The images were centered in a 28 × 28 image by computing the pixel’s center of mass and translating the image to place this point in the 28 × 28 field’s center (Lecun et al., 1998). From the large MNIST database, 500 samples were chosen randomly. At each of the five-fold cross-validation runs, 400 data points served as training points, and 100 data points served as testing points. Note that for this dataset, each data point which is represented by a 28×28 matrix was transformed into a 1×2025 vector using histogram of oriented gradients, a feature descriptor used in computer vision and image processing.

Tables 12–15 show the performance of PHCA and the five major classification algorithms in terms of precision, recall, specificity, and F1-score per class, and accuracy. Barplots of these results are presented in Fig. 9. PHCA obtained an accuracy of 92%. It was outperformed by RF and SVM with accuracy of 95 and 93, respectively. Moreover, Table 16, which is a summary of the Nemenyi post-hoc test results, show that the performance of PHCA is not significantly different from the performance of the other classification algorithms, except from the performance of CART. The mean rank difference of 51.469709 between CART and PHCA, and the p−value of 0.01919 show that PHCA outperformed CART significantly for the sampled MNIST data points, in terms of the mean performance metrics.

Table 12 Precision per class of each of the six methods when classifying the MNIST dataset.

Classifier	Precision per class (%)	
	1	2	3	4	5	6	7	8	9	10	
LDA	90.9	88.9	95.7	92.2	94.1	93.5	97.9	93.2	86.8	88.7	
CART	77.8	50.0	NA	23.8	NA	NA	57.1	26.7	NA	NA	
KNN	90.9	81.8	90.9	90.9	100.0	100.0	90.9	100.0	100.0	83.3	
SVM	100.0	90.0	76.9	100.0	90.0	100.0	100.0	87.5	100.0	88.9	
RF	90.9	90.0	90.9	100.0	100.0	100.0	100.0	100.0	100.0	83.3	
PHCA	100.0	92.0	92.0	96.0	94.0	86.0	96.0	86.0	86.0	92.0	

Table 13 Recall per class of each of the six methods when classifying the MNIST dataset.

Classifier	Recall per class (%)	
LDA	100.0	96.0	88.0	94.0	96.0	86.0	92.0	82.0	92.0	94.0	
CART	70.0	60.0	0.0	100.0	0.0	0.0	40.0	80.0	0.0	0.0	
KNN	100.0	90.0	100.0	100.0	80.0	90.0	100.0	70.0	90.0	100.0	
SVM	100.0	90.0	100.0	100.0	90.0	100.0	100.0	70.0	100.0	80.0	
RF	100.0	90.0	100.0	100.0	80.0	100.0	100.0	80.0	100.0	100.0	
PHCA	92.6	88.5	95.8	84.1	100.0	87.8	92.3	97.7	95.6	79.3	

Table 14 Specificity per class of each of the six methods when classifying the MNIST dataset.

Classifier	Specificity per class (%)	
	1	2	3	4	5	6	7	8	9	10	
LDA	100.0	99.6	98.7	99.3	99.6	98.5	99.1	98.0	99.1	99.3	
CART	97.8	93.3	100.0	64.4	100.0	100.0	96.7	75.6	100.0	100.0	
KNN	98.9	97.8	98.9	98.9	100.0	100.0	98.9	100.0	100.0	97.8	
SVM	100.0	98.9	96.7	100.0	98.9	100.0	100.0	98.9	100.0	98.9	
RF	98.9	98.9	98.9	100.0	100.0	100.0	100.0	100.0	100.0	97.8	
PHCA	100.0	99.1	99.1	99.6	99.3	98.5	99.6	98.5	98.5	99.1	

Table 15 F1 score per class and accuracy of each of the six methods when classifying the MNIST dataset.

Classifier	Acc (%)	F1 score per class (%)	
		1	2	3	4	5	6	7	8	9	10	
LDA	92.0	95.2	92.3	91.7	93.1	95.0	89.6	94.8	87.2	89.3	91.2	
CART	35.0	73.7	54.5	NA	38.5	NA	NA	47.1	40.0	NA	NA	
KNN	92.0	95.2	85.7	95.2	95.2	88.9	94.7	95.2	82.4	94.7	90.9	
SVM	93.0	100.0	90.0	87.0	100.0	90.0	100.0	100.0	77.8	100.0	84.2	
RF	95.0	95.2	90.0	95.2	100.0	88.9	100.0	100.0	88.9	100.0	90.9	
PHCA	92.0	96.2	90.2	93.9	95.0	96.9	86.9	94.1	91.5	90.5	85.2	
Number of data points:	500		Number of classes:	10	
Training set size:	400		Number of attributes:	2,025	
Testing set size:	100								

Figure 9 Barplots of performance metrics of PHCA and the five other classifiers for the MNIST dataset.

Table 16 Nemenyi’s test of pairwise comparison of the different classification algorithms for the MNIST dataset.

	Mean rank difference	p-value	
CART-LDA	−48.59166	0.03322	
KNN-LDA	16.54878	0.88258	
SVM-LDA	32.07317	0.27317	
RF-LDA	43.57317	0.04462	
PHCA-LDA	2.87805	0.99997	
KNN-CART	65.14044	0.00086	
SVM-CART	80.66483	1.0e−05	
RF-CART	92.16483	2.1e−07	
PHCA-CART	51.46971	0.01919	
SVM-KNN	15.52439	0.90814	
RF-KNN	27.02439	0.47074	
PHCA-KNN	−13.67073	0.94496	
RF-SVM	11.50000	0.97367	
PHCA-SVM	−29.19512	0.37988	
PHCA-RF	−40.69512	0.07541	

The five validation experiments exhibit that the proposed classifier, PHCA, can be used to solve binary or multi-class classification problems, classification problems involving points from the n-dimensional euclidean space or image data, classical or synthetic datasets, and datasets whose number of features ranges from two to the thousands. Moreover, the performance of PHCA has no significant difference form the performance of the benchmark classifiers, except for a few instances where a benchmark classification algorithm performed significantly poorer than the performance of PHCA.

These validation results do not imply that PHCA is better than any of the other major classification algorithms. However, these results illustrate the no-free lunch theorem, which implies that no learning algorithm works best on all given problems. Moreover, these results suggest that PHCA can be at par or even better than some classifiers in solving some particular classification problems.

What sets PHCA apart from the well-known machine learning classifiers is that it is non-parametric, but at the same time a linear classifier. It is a non-parametric algorithm in the sense that it does not restrict the data to follow a particular distribution, nor fix the number of datasets’ parameters for the algorithm to work.

Conclusions

The main contribution of this study was the development of PHCA, a non-parametric but linear classifier which utilizes persistent homology, a major and very powerful TDA tool. PHCA was applied in solving four different classification problems with varying sizes, number of classes, and number of attributes. For the four classification problems, the performance of PHCA was measured and compared to the performance of LDA, CART, KNN, SVM, and RF, in terms of precision, recall, specificity, accuracy, and F1 score. A five-fold cross-validation was used in all validation runs. PHCA performed impressively in each of the four classification problems. PHCA ranked either second or third in the first three datasets in almost all metrics; although it ranked 5th in the synthetic dataset, it obtained an accuracy of 99.16%. Additionally, all the classifiers, except for PHCA, ranked last in terms of accuracy and F1 scores in at least one of the four classification problems. In conclusion, the validation results show that PHCA can perform well, or even better, than some of the widely used machine learning classifiers in solving classification problems. Moreover, this study does not imply that PHCA works better than other machine learning algorithms, but this shows that PHCA can work in solving some classification problems.

The validation of PHCA in this study was limited to relatively small problems which are restricted by the computers used in this study. PHCA can be further validated by considering larger problems and by using more powerful computers which can solve problems involving datasets with higher dimensions. Furthermore, some improvements that can be imposed on the proposed classification algorithm in this study is by considering other topological signatures or by considering PH representations other than persistence diagrams, such as persistence landscapes. Recent advancements and modifications on the computation of persistent homology may also be implemented to possibly improve the performance of PHCA. Validation of the proposed algorithm were implemented in R using TDA package and GUDHI library in solving persistent homology of data. It should be noted that there are other platforms and solvers which can be used, like JavaPlex, Perseus, Dipha, Dionysus, jHoles, Rivet, Ripser, and PHAT, which offer some variations in the way PH can be computed. Indeed, this study has opened many of research opportunities which can be explored by mathematicians, data scientists, and computer programmers.

Supplemental Information

Supplemental Information 1 The program code written in R used to analyze Dataset 1 (Iris Dataset).

Click here for additional data file.

Supplemental Information 2 The program code written in R used to analyze Dataset 2 (Wheat Seeds Dataset).

Click here for additional data file.

Supplemental Information 3 The program code written in R used to analyze Dataset 3 (Social Network Ads Dataset).

Click here for additional data file.

Supplemental Information 4 The program code written in R used to analyze Dataset 4 (Synthetic Dataset).

Click here for additional data file.

Supplemental Information 5 The program code written in R used to analyze Dataset 5 (MNIST).

Click here for additional data file.

Supplemental Information 6 The Iris plant dataset which was used as the first validation dataset is available at the UCI Machine Learning Repository: https://archive.ics.uci.edu/ml/datasets/iris.

Click here for additional data file.

Supplemental Information 7 The wheat seeds dataset which was used as the second validation dataset is available at the UCI Machine Learning Repository: https://archive.ics.uci.edu/ml/datasets/seeds.

Click here for additional data file.

Supplemental Information 8 The social network ads dataset which was used as the third validation dataset is available from Kaggle: https://www.kaggle.com/rakeshrau/social-network-ads/version/1.

Click here for additional data file.

Supplemental Information 9 The synthetic dataset is the fourth validation dataset and this dataset can be generated when ALLvsPHCA_to_solve_4Synthetic.R is run in R.

Click here for additional data file.

Supplemental Information 10 The MNIST dataset is the fifth validation dataset and this dataset can be generated when ALLvsPHCA_to_solve_4Synthetic.R is run in R.

Click here for additional data file.

Supplemental Information 11 Background information about persistent homology.

Click here for additional data file.

This work was created with the guidance of Dr. Clarisson Rizzie P. Canlubo of the University of the Philippines Los Baños and Dr. Rachelle R. Sambayan of the University of the Philippines Diliman.

Additional Information and Declarations

Competing Interests

Author Contributions

Data Availability

The authors declare that they have no competing interests.

Mark Lexter D. De Lara conceived and designed the experiments, performed the experiments, analyzed the data, performed the computation work, prepared figures and/or tables, authored or reviewed drafts of the article, and approved the final draft.

The following information was supplied regarding data availability:

The program codes and datasets used in the algorithm validation is available at GitHub: (https://github.com/mlddelara/PHCA) and at Zenodo (https://doi.org/10.5281/zenodo.7359282); Kramer, Bianca, & Bosman, Jeroen. (2019). Open access potential and uptake in the context of Plan S—a partial gap analysis. Zenodo. https://doi.org/10.5281/zenodo.3543000.

The data and codes are also available in the Supplemental Files.

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
