# Peer review of "Persistent homology classification algorithm"

_PeerJ Computer Science, doi:10.7717/peerj-cs.1195_

## Round 0.1 · original submission · Major Revisions

Please consider all the suggestions given by the reviewers, especially those related to the improvement of the style and English grammar. Pay attention to the suggestions given by reviewer 2 about the experimental design.

·

Basic reporting

The topic of the paper is important and up-to-date. The article was not organized into sections and subsections for easy readability. The authors fail to provide detailed review of related works in the area, The article could be improved in this regard.

On Page 3-4, Line 139-185, Poor description of the procedure for training and testing, difficult for readers to understand what the author is trying to communicate. The author should work toward fixing some of the grammatical and punctuation errors contained in the manuscript. Suggestions on how to fix some these errors are provided in the addition comments section.

Experimental design

The author elaborated the methodology and discussed the research findings in light of important mathematical and machine learning concepts. However, the authors should work toward fixing some of the grammatical and punctuation errors contained in the manuscript.

The following are some major comments that could help the author to improve the manuscript:

1. Page 3, 120-121, what is the justification for choosing the split method with 80% training set and 20% for testing. Why did you choose not to use the cross validation method?

2. The author should perform significance tests (Nemenyi or any suitable statistical test) for the results in Table 1, 2, 3 and 4.

3. The author should provide graphs (figures) for the results in Table 1-4 for better visualizations of the differences in performance for the proposed method and the baseline methods. These graphs/figures could be based on the accuracies obtained on the proposed method in comparison with the five baseline methods.

Validity of the findings

In my opinion, the novelty in the findings/work is weak. The author claimed to propose a new supervised classification algorithm (as claimed on line 15-17, 88-89, 110-112 and 196) but failed to conduct detailed literature review on the use persistent homology and its application in machine learning as contained in some of the following articles

1. Edelsbrunner, H., & Harer, J. (2008). Persistent homology-a survey. Contemporary mathematics, 453, 257-282.
2. Pun, C. S., Xia, K., & Lee, S. X. (2018). Persistent-Homology-based Machine Learning and its Applications--A Survey. arXiv preprint arXiv:1811.00252.

The proposed algorithm was evaluated/validated using three (3) classical datasets and one (1) syntactic dataset and its performance compared with five baseline classification algorithms. Program codes for the experiments implemented in R programming language were also provided via GitHub.

Additional comments

Minor comments:

- Page 1, line 10 – But none of the algorithms.... should be..... but none of the classifiers.
- Page 1, line 23 – that can automatically… instead of using … which has the ability...
- Page 1, line 37 – … have found may application….. have found many applications
- Page 1, line 35 – … also called classifier….. should be also called a classifier
- Page 1, line 41 – .....is a classification application should be..... is a classification task
- Page 2, line 47 – classification algorithms……. several of these techniques, the author should be
consistent on the use of terms.
- Page 2, line 67 – Provide reference for the topological data analysis (TDA)
- Page 2, line 73 – well understood should be well-known
- Page 2, line 78 and 84– consider revising lines 78 and 84
- Page 2, line 90- …Published works about the fusion of these topics are quite new… should be ….
are quite few.
- Page 2, line 92 – … machine learning algorithms and its applications should be....machine learning
algorithms and their applications
- Page 2, line 97- While in this study... should read..... In this study, we have used topological….
- Page 3, line 100… has been gaining attraction ... should read.....has been gaining attention
- Page 3, line 114… is to divide …….and testing... should read..... and testing set
- Number all the equations in the manuscript.
- The figures (1-11) for the persistent diagram and persistent barcode are not very clear for easy
understanding by readers.
- Use of peristence instead of persistence in all Figures (1-11).

Reviewer 2 ·

Basic reporting

The writing could be improved significantly. The introduction could be re-written to focus on aspects of the machine learning paradigm directly related to the proposed algorithm. In its current form, the introduction is trying to do too much by attempting to discuss too many details about classification and machine learning, and since this is done in broad strokes, I am not sure that it adds significantly to the paper. Perhaps the author could focus on how the proposed algorithm bridges the gap between classical classifiers and machine learning-based classifiers, and how the proposed algorithm builds on this approach. The flow and cohesion among the paragraphs could also be improved. Perhaps the author could partition the introduction into a few sections, each focusing on a specific topic? Grammatical errors abound throughout the text.

The literature discussed is far from complete. The main literature that the author cites on persistent homology-based machine learning is very recent. Earlier papers abound on machine learning approaches that tap persistent homology features to perform a variety of tasks --- image classification, signal processing, analysis of neural connections, and manifold learning to name a few. I leave it to the author to discover these on their own. Also, while the author compares the proposed approach to classical and other established classifiers, there was no mention of other persistent homology-based classifiers that go beyond feature engineering and have been around for some time now. The author should definitely include discussions on these to provide more contextualised comparison.

The data sets used to test the proposed algorithm are on the "traditional" side. Since these are already well-understood, perhaps the author could include benchmarking on more current go-to data sets for classification tasks (MNIST, CIFAR, etc.) The author should also consider including illustrations to help develop intuition on the proposed algorithm.

While a detailed account of the algorithm is provided, no formal theoretical result was presented to support the algorithm.

Experimental design

The topic of the manuscript falls within the scope of the journal and the objective of the manuscript is clearly stated. However, in its current form, I am not convinced that the manuscript offers a substantial novel contribution to fill an identified gap in the literature. In my opinion, the manuscript fails to position the proposed algorithm as a necessary contribution as it simply provides another example of a classifier that uses engineered features from computed topological signatures.

There also seems to be a confusion on what should actually be included in the algorithm. For example, splitting the data set for training and testing is a standard step to evaluate classifiers. Unless I am misunderstanding something in the current presentation of the manuscript, this should be excluded in the algorithm itself. I understand that the main idea of the proposed algorithm is to use the intrinsic topological information within a known cluster to see how a new point changes a known state --- this part is novel. Since any classifier needs to be trained on some prior data, I am confused about the need to include the train/test split in the algorithm itself.

I am also unsure about the use of "algorithm" to describe the approach. The main basis of the classification seems to be due to a score provided by a multivariate model that uses features engineered from topological signatures. The model needs to be more prominently and rigorously discussed in the manuscript. In particular, more details should be included regarding

(i) what, why and how variable features are selected and processed;
(ii) the model design as a linear aggregate;
(iii) the equal weights among the variables (were the variables scaled?);
(iv) why persistence diagrams are preferred over landscapes;
(v) why the Wasserstein distance (and how p=2 was chosen) was preferred over the bottleneck distance.

As it is, there is not enough information to properly appreciate the basis of the multivariate model. Were more variables considered but dropped due to multicollinearity/dependence? I have no problem calling this model a classifier, but I am unsure about referring to it as an algorithm. Again, I may be misunderstanding a critical portion of the manuscript --- which may hint that something must be rewritten to avoid confusion.

Finally, it would be nice if the author can explore the direct integration to the proposed classification algorithm of other novel ideas used in the theory of persistence, or even classical homology in a manner that goes beyond simply appealing to the engineering of features from extracted topological information.

Validity of the findings

The computational results on classification accuracy seem valid, although somewhat expected. At the very least, the proposed method seems to perform at par with a variety of classifiers. However, as stated above, the data sets used in the tests are dated, and very small considering that one goal is to eventually use the algorithm for machine learning where data sizes are several orders of magnitude larger. In this context, the results seem less exciting.

One noticeable missing component in the benchmarking is the performance of other persistent homology-based algorithms already available in the literature. Without this, it is hard to appreciate the significance of the paper's contribution, which, in my opinion, is what would merit its publication.

Other standard results such as complexity analysis (run time and memory costs) should be discussed and substantiated with relevant computational results.

Additional comments

Overall, I think the approach has potential if explored rigorously, and demonstrated to work on data sets at scale.

---

## Round 0.2 · Major Revisions

Please pay attention to the comments provided by the reviewers, especially reviewer two's comments.

·

Basic reporting

The grammar used in the first manuscript has been greatly improved especially with regards to literature review and references. However, the article can still be improved in the following areas:

1. Page 4 (Line 145-150) is too scanty to explain the proposed classifier
2. Page 6 (Line 225)…. What follows is the pseudocode. Is Algorithm 1 a pseudocode?
3. The author should use better graph tools like Gnuplots and ggplot2 for better visualization instead of using excel charts?

Experimental design

Most of the issues raised during the first review circle like the justification for using split method instead of cross-validation to evalaute the model has been addressed. But I am still of the opinion that the following can improve the manuscript:

1. There should be a table (suggested to be Table 8) to compare the accuracies obtained using the five baseline methods and the proposed method for all the datasets. Graph/figure could be based on the accuracies obtained on the proposed method in comparison with the five baseline methods).

2.Statistical significant test (like Nemenyi Test) should be performed using Table 8.

Validity of the findings

no comments

Additional comments

1. Page 2-3, Check that all the acronym on page 2-3 are defined (provide full meaning of all acronyms during the first usage)
2. Page 3, line 108……. and showing that better…… This needs to be revised
3. Page 3-4, line 131-138- The paragraph does not make any sense and it needs to be revised .
4. Page 6, line 223…. The score function, score (yj) is computed as:
5. Headings (Tables name and description) for Table-1-6 should be on top of the table
6. The scores in Table 1-6 should be in two decimal places for uniformity

Reviewer 2 ·

Basic reporting

The writing of the manuscript presents a significant improvement from the version previously reviewed by this referee. The author did a good job at implementing most of the suggestions to improve the flow and readability of the manuscript. Although the presentation of ideas and sentence construction tend to be on the basic side (and somewhat fragmented), the technical preciseness and breadth of topics expounded are at an appropriate level. Some grammatical errors (mostly on proper verb use) are still present throughout the manuscript. Some figures could have been given more utility. For example, Figure 1 was only used to illustrate the filtration process and was never referred to again. One way to give more purpose to this particular figure would have been to superimpose it with the resulting persistence barcode to demonstrate how the detected persistent features relate to the evolution of the simplicial complex.

Although the author did a splendid job at conducting a thorough literature search, the offered explanation on how the proposed algorithm differs from other classifiers that use persistent homology did not help me appreciate it (quite frankly, it sounds too convenient). The offered explanation seems to distinguish classification from clustering. Why this was relevant to the task at hand wasn’t, at least to me, very clear yet. It did, however, help clear up a bit why the proposed algorithm would not be applicable to the MNIST or CIFAR data sets which I suggested. I think it would have helped more if the author focused on highlighting the differences in the actual unique structure/primary idea between the classifiers. The author can then use this as a jump off point to discuss where the algorithm can be appropriately applied to. At the very least, the central idea of harnessing changes between persistence diagrams (before and after a new point is added) should be prominently discussed in this section as it is the primary novel idea of the algorithm.

In view of the response regarding the use of other up-to-date go-to data sets for classification, I would have appreciated it if the author took the initiative to find other appropriate data sets other than MNIST or CIFAR that would demonstrate the proposed algorithm’s utility. My comment still stands that, while the use of classical data sets does not take away from presented computational results, they seem less exciting as the data sets used to demonstrate utility are already well understood. This also relates to the statement made by the author that the algorithm is designed to handle “datasets with lower dimension that cannot be separated by hyperplanes” --- if the low dimensional data can already be separated by hyperplanes, then where is the challenge in the classification?

Finally, the author did not respond to, nor address the last point I made in this section that no formal theoretical guarantee was presented to support the proposed algorithm.

Experimental design

Again, the author did splendid work in restructuring the relevant sections based on most of the earlier provided comments and suggestions. The presentation of the classifier is now better, with the main idea highlighted appropriately. I suggest adding another figure right after the opening paragraph of the section to illustrate the general pipeline and develop intuition about the algorithm.

With respect to my comments on the scoring function, the author provided additional discussions on

1. What, why and how variable features are selected and processed. The resulting parameters in the scoring function seem to work, at least computationally. A critical gap in this development, however, is the theoretical guarantee on the primary observation which was the basis for almost all parameters --- that adding a new point in a known cluster results to the claimed changes in the topological signatures detected by persistent homology. For now, I will refer to this as the author’s “primary hypothesis”. As there are many possible cases that need to be checked to verify this, some overarching general theoretical guarantee is necessary. As it is, the proposed algorithm stands on an unsupported hypothesis on topological changes within clusters when a new point (to be classified) is added. This has direct implications on the merits of the scoring function.

2. The model as a linear aggregate. The author has not discussed any particular reason as to why a linear model is preferred. My appreciation is that the author simply wanted to test whether this approach would work, and never looked back when some evidence showed that it seemed to do so.

3. The equal weights among the variables. The author discussed the choice for the sign (positive or negative) based on the primary hypothesis. Why the coefficient was made to be only either +1 or -1 is still unexplained. This is critical since the parameters are measured in different orders of magnitude.

4. Why the Wasserstein distance was preferred over the bottleneck distance. This was sufficiently explained.

The discussion of the proposed algorithm’s complexity --- that it performs in “polynomial time” --- is hand-waving at best and does not offer particularly useful insights. To me, this hints that the complexity of the algorithm being proposed has not been fully considered by the proponent. At the very least, computational evidence supporting the claim of polynomial complexity needs to be provided.

Finally, the presented evaluation of the proposed algorithm demonstrates its performance relative to classical classifiers. What it does not achieve, however, is positioning --- whether or not it performs better overall, or in particular types of datasets --- the proposed algorithm with respect to available classifiers in its category (those that tap into persistent homology). In my opinion, this is very important as it justifies the need for the proposed algorithm and demonstrates the gap it fills and advantages it offers.

Validity of the findings

As before, the results seem valid but expected. The implementation of cross-validation is a welcome improvement and adds to the validity of the results. However, as there was no other significant change in the experiments performed, my earlier comments still stand.

Also, I think the author is missing the point --- the issue is not with how the proposed algorithm is different from other available classifiers, but rather with how well it performs relative to them. If the algorithms can be applied to the same data, then there is no reason why benchmarking cannot be performed.

Additional comments

Overall, I still am not convinced on the absolute necessity of the proposed algorithm primarily due to the following:

1. Lack of theoretical guarantees on critical points about the scoring function and complexity;

2. It is hard to appreciate the computational results without appropriate benchmarking, i.e. with other classifiers in its category (those that tap into persistent homology). For example, without this benchmarking, it is hard to provide a fair comparison between the proposed algorithm and CBN (this is one of the algorithms in the same category that the author cited). However, as CBN is unsupervised, one may argue that it has clear advantages over the proposed algorithm. Benchmarking their performances on the same data sets can provide clarity on this argument.

3. The presented computational results are with respect to very small data sets that are already well understood.

---

## Round 0.3 · Minor Revisions

Please follow the recommendations given for this new version of the paper.

·

Basic reporting

The grammar used in the latest manuscript has been greatly improved especially with
regards to literature review and references. However, the article can still be
improved in the following areas:

• Page 2 (line 48): The no-free lunch theorems. change theorems to theorem
• Page 3 (line 101): on definition 3 ‘Ks is’ is repeated
• Page 5 (line 163), page 9 (line 321) and page 10 (line 362) … in Section. What section?
• Page 8-9, Acronyms are not defined
• Page 13 (line 430) … will be classified under a class if its. Change ‘its’ to it’s
• Page 18 (line 474): … significantly better that KNN. Change ‘that’ to ‘than’.
• Page 11 (Line 373)…. What follows is the pseudocode. Is Algorithm 1 a pseudocode?

Experimental design

Most of the issues raised during the first review circle like the justification for using
split method instead of cross-validation to evalaute the model has been addressed.

Validity of the findings

I advise that the author carefully go through all the values on the tables and the figures in order to verify claims made in the article are true in all cases. However, I have the following comments that I want the author to address:

• Page 13 (line 426) PHCA should not be included in the five benchmark classification algorithms.
• Page 14 (Figure 5): The figures don’t depict what's exactly on the tables.
• Page 16 (Figure 6): The figures doesn't represent the tables exactly.
• Page 17 (line 457): … which is at 90.95%. Change 90.95% to 90.45%.
• Page 17 (line 459): Highest precision for class 3 and highest specificity for class 3 are not true claims.
• Page 17 (line 459): …recall for class 1. Change to class 3
• Page 18 (figure 7): The figures does not represent the tables exactly.
• Page 18 (line 469 - 470): The claim is not true according to the tables.
• Page 18 (line 475 - 477): The claim should be revised.

---

## Round 0.4 · accepted · Accept

Reviewers consider the paper is ready for publication, and so do I.

·

Basic reporting

The grammar used in the latest manuscript has been greatly improved especially with
regards to literature review and references. However, the only minor concern I raised from my previous review comment that was not addressed by the author is

• Page 5 (line 163) To do this we follow the construction presented in Section. What section?

Experimental design

Most of the issues raised during the first and second review circle like the justification for using
split method instead of cross-validation to evaluate the model has been addressed.

Validity of the findings

I suggested to the author in my last review cycle to carefully go through all the values on the tables and the figures in order to verify claims made in the article are true in all cases. I have checked and comfirm the validity of the finding. Moreover, during the first review circle, I suggested to the author to perform some statistical significance test which the author did Nemenyi Test.

Additional comments

I have no additional comments because most of what I raised during the review cycles has been addressed by the author.